# Base Pairing Promoted the Self-Organization of Genetic Coding, Catalysis, and Free-Energy Transduction

**DOI:** 10.3390/life14020199

**Published:** 2024-01-30

**Authors:** Charles W. Carter

**Affiliations:** Department of Biochemistry and Biophysics, University of North Carolina at Chapel Hill, Chapel Hill, NC 27599-7260, USA; carter@med.unc.edu

**Keywords:** aminoacyl-tRNA synthetase•tRNA cognate pairs, bidirectional genetic coding, protein folding, AND gating, origin of catalysis, origin of free-energy transduction, genome propagation into the proteome, phylogenetics, ancestral gene reconstruction, selection constraint surface, reciprocally coupled gating

## Abstract

How Nature discovered genetic coding is a largely ignored question, yet the answer is key to explaining the transition from biochemical building blocks to life. Other, related puzzles also fall inside the aegis enclosing the codes themselves. The peptide bond is unstable with respect to hydrolysis. So, it requires some form of chemical free energy to drive it. Amino acid activation and acyl transfer are also slow and must be catalyzed. All living things must thus also convert free energy and synchronize cellular chemistry. Most importantly, functional proteins occupy only small, isolated regions of sequence space. Nature evolved heritable symbolic data processing to seek out and use those sequences. That system has three parts: a memory of how amino acids behave in solution and inside proteins, a set of code keys to access that memory, and a scoring function. The code keys themselves are the genes for cognate pairs of tRNA and aminoacyl-tRNA synthetases, AARSs. The scoring function is the enzymatic specificity constant, k_cat_/k_M_, which measures both catalysis and specificity. The work described here deepens the evidence for and understanding of an unexpected consequence of ancestral bidirectional coding. Secondary structures occur in approximately the same places within antiparallel alignments of their gene products. However, the polar amino acids that define the molecular surface of one are reflected into core-defining non-polar side chains on the other. Proteins translated from base-paired coding strands fold up inside out. Bidirectional genes thus project an inverted structural duality into the proteome. I review how experimental data root the scoring functions responsible for the origins of coding and catalyzed activation of unfavorable chemical reactions in that duality.

## 1. Introduction

Watson and Crick’s model for DNA structure [1,2] may be the most decisive dividing line in the history of human awareness. Nucleotide base pairing revealed the molecular basis of inheritance, redirecting biological and human health research in a single stroke, from inspired—but blind—guesswork to lucid inevitability. Inheritance, with variation, is only one of several pillars underpinning living matter. Sentience as we know it also requires molecular processes that store and manipulate both information and free energy. Biology is a singularly self-constructing form of matter.

Interpreting information stored in genes required the symbolic transformation embodied in the universal genetic coding table. That table pairs amino acids to one or more of the 64 triplet “codons” possible using the four nucleotide bases found in nucleic acid genes. That such a code existed [3,4,5], what the assignments were [6,7], and the nature of the assignment catalysts [8,9] all emerged rapidly after Watson and Crick described the DNA double helix. It was roughly 30 more years until the first clues [10,11,12,13] began to emerge that would lead, ultimately, to the surprising conclusion that consequences of base pairing might also propagate into the proteome itself and be central to the emergence of both the coding table and its molecular implementation [14].

How Nature discovered genetic coding is a largely ignored question, yet the answer is key to explaining the transition from biochemical building blocks to life. Genetic coding also conceals a deeper, rarely stated question. The molecular assignment catalysts, aminoacyl-tRNA synthetases (AARSs), must implement the very language in which their own genes are written. That means that the AARSs are reflexive [15,16,17]. Reflexivity sets living matter apart from all other forms of active matter. Its roots lie deep in evolutionary molecular biology, which must have resulted from a complex historical progression. Each step in that historical progression exploited only what was available at the time and must have enabled the next step. Solving the puzzle of genetic coding means charting that process.

The proteome amplifies the chemical engineering diversity embedded in genes by perhaps a billion-fold [15]. That amplification introduced computational control enabling life to emerge and flourish on Earth. Nature enabled it by evolving the genetic code. That reagent-to-token assignment is performed by nanomachines—(AARS)•tRNA cognate pairs. As their name suggests, AARSs use ATP to activate and transfer the α-carboxyl group of amino acids covalently to cognate tRNAs, thus enforcing the code.

Nature evolved these self-constructing machines within an as-yet unknown historical context. Pre-existing conditions, illustrated by [18,19,20,21,22,23,24] made each successive step possible. Each step, in turn, enabled those that followed. Creating rudimentary assignment catalysts, in turn, enabled the explosive transition to living organisms by a process we have compared to booting a computer’s operating system [15]. The code was almost entirely completed before the Last Universal Common Ancestor, LUCA [25]. Hence, most aspects of the process must have been highly cooperative. Cooperativity, in turn, meant that many unlikely processes had to promote one another, increasing their joint probability.

The dashed curves in Figure 1a suggest four separate areas in which evolutionary advances must have promoted one another as the coding alphabet grew. We imagine that the coding alphabet was initially modest, perhaps only a single bit. Acquiring new bits required the AARSs to speciate, in order to better discriminate between amino acid and tRNA substrates. That meant that mutant sequences folded into structures with more sophisticated cognition (Figure 1a). The coding alphabet size, specificity, folding, and function (Figure 1a) all have experimentally accessible signatures (Figure 1b). These can help us track the development of the code, using experimental models of evolutionary intermediate AARS•tRNA cognate pairs.

The interdependence evident in Figure 1a also orders events in time. Models for the origin of coding must thus identify plausible pathways between successive steps. Processes that favor one another confer a selective advantage and tend to survive. Our studies suggest that catalysis itself was prominent even in the earliest ancestral models [2]. Moreover, peptide bond formation is an unfavorable reaction that must be driven by coupling it to a source of free energy. So, the earliest catalysts must also have coupled chemical free energy into biological reactions, as we have observed [2]. Specificity, however, required lengthy refining of binding specificities to enhance precision. It came only later.

How did Nature solve various “chicken-and-egg” questions by converting random chemistry into symbolic coding to interpret genes? This question is especially vexing when we consider genes whose translated products enforced the coding rules. How too did Nature learn to exploit sources of chemical free energy necessary to sustain itself far from equilibrium? Nearly 4 billion years later, these questions remain deeply mysterious to us—the only products of that explosive transformation who are able to ask them.

My colleague, Peter Wills, and I re-focused attention [15] to a specialized subset of questions. We considered only a narrow range of phenomena with two characteristics. First, we examined only areas where the relevant fields still have useful experimental signals. Second, we looked for the elements of reflexivity—specific recognition of amino acid and tRNA substrates—in models for the primordial AARS assignment catalysts.

Coherent shards from evolutionary molecular biology, structural biology, bioinformatics, and biochemistry are beginning to suggest useful answers. These come from a wide range of experimental and conceptual disciplines. The discussion will proceed as follows. Section 2 summarizes the key model for experimental study of the origin of the genetic coding table and its implementation. Section 3 asks what can be expected from an analysis of the historical record. Section 4 describes the constraint surfaceon which the scoring function probably shaped Nature’s self-organization prior to the advent of Darwinian selection. Section 5 surveys challenges that remain to be addressed and the experimental models with which to address them. Section 6 considers whether understanding the emergence of animate matter required new physics.

## 2. Creating the First Bit of Genetic Information from Scratch

At the boundary between non-random chemistry and biology, the earliest AARS•tRNA cognate pairs began to embed two distinct kinds of symbolic information into nucleic acids. Transfer RNAs are like a computer programming language connecting amino acid physical chemistry to symbols, i.e., the 64 codons. Messenger RNAs are blueprints for making functional proteins written using that language [27]. Together, they separated phenotype from genotype. Unlike the Morse code, which was assembled to represent a pre-existing alphabet, Nature created alphabet, symbols, and programs simultaneously and from scratch.

Postulating the necessity of a “first bit” appears to contradict prevailing work on the origin of coding [28]. Those authors described a computer simulation in which competing “protocodes” gradually settled on an optimal assignment of amino acids to codons. However, the contradiction is only apparent. The most important requirement for a functional coding system capable of subsequent refinement and expansion is that it produce binary patterns in the coded polypeptides consistent with folding into secondary structures—α-helices, β-strands, and turns. Any rudimentary code would necessarily be only marginally specific and thus would code for quasispecies. Elements described in this section satisfy this requirement to an extraordinary extent.

### 2.1. AARS/tRNA Cognate Pairs Function as Mutually Exclusive Molecular AND Gates

The elemental barrier to creating the code was creating the first bit of information. AARS “assignment catalysis”—selecting and combining two substrates from closely related homologs—is a molecular homolog of AND gating in computer hardware. The first genetic coding “bit” required a rudimentary molecular AND gate. That functionality likely was defined by two, mutually exclusive AARS•tRNA cognate pairs (Figure 2). Both AARSs and tRNAs use separate domains for catalysis and anticodon recognition. AARSs still recognize the acceptor stem in “minihelices” lacking the anticodon-binding domain derived from tRNAs for several amino acids [12].

De Duve [29] pointed out that AARSs likely used a distinct set of molecular recognition codes for cognate tRNAs. He called the tRNA bases recognized by synthetases paracodons. Schimmel et al. [12] proposed that these paracodons likely resided in the acceptor stems and referred to them as an “operational” code. That code likely drove recognition in single-domain AARS•tRNA complexes prior to the advent of the anticodon stem-loop. We outline in Section 2.6 how the earliest cognate pairs we have characterized could have discriminated between both Class I and II amino acid and TΨC minihelix substrates. An important hypothesis is that this molecular AND coding is rooted in bidirectional coding of Class I and II AARSs on opposite strands of the same ancestral gene.

### 2.2. Bidirectional Genetic Coding Projected Duality into the Proteome

The molecular basis of this rudimentary substrate differentiation surfaced with the recognition that AARSs have two distinct versions, Classes I and II [10,30,31]. The two Classes have different architectures. Class I active-site domains are modified Rossman dinucleotide-binding folds with parallel β-strands interspersed with α-helices. Class II AARSs have multi-stranded antiparallel β-sheets [32,33,34]. The Class distinction also rationalized earlier studies showing that, with the notable exception of aromatic amino acids, those AARSs now belonging to Class I acylated the 3′terminal ribose 2′ OH; those belonging to Class II acylated the 3′ OH. Ribas and Schimmel [11] later summarized subclass-specific pairwise interactions matching Class I AARSs with the minor groove and Class II AARSs with the major groove of cognate tRNAs (Figure 2a). Class I and II AARSs might have stabilized all cognate tRNAs by binding as envisioned in Figure 2a. Note, however, that only the cognate interaction (blue vs. green) leads to functional aminoacylation.

Despite their contrasting architectures, Class I and II AARSs appear to share a common origin. Rodin and Ohno [13] observed unusually high base pairing between the antiparallel-aligned coding sequences of the highly conserved PxxxxHIGH and KMSKS signatures of Class I AARSs with Motifs 2 and 1, respectively, in Class II AARSs. They proposed that the original Class I and Class II ancestors were coded by opposite strands of the same bidirectional gene. Their proposal received little attention for another decade. Subsequent work, however, Refs. [2,35,36,37,38,39,40] provided substantial supporting evidence.

The base-paired segments represent only ~130 residues, about 40% of the smallest Class I AARSs (TrpRS). Remarkably, those segments include the active sites of both Class I and II AARSs. When purified, they retain ~60% of their activity in both key reactions [2,39,41,42,43]. We called them AARS “urzymes”. Improbably elevated middle-base pairing between signature sequences extends throughout >75% of the paired, antiparallel coding sequences [35]. Moreover, the base pairing between sequences independently reconstructed for Class I and II urzymes increases toward the oldest ancestral node [44].

Class I AARSs all share the Rossmann dinucleotide-binding fold containing the active site. Evolution has added two distinct additional domains: (i) a C-terminal anticodon-binding domain and (ii) a variable length insertion known as a connecting peptide (dashed line in Figure 2c). Class II AARSs catalytic domains are based on an extended antiparallel β-sheet. They also contain a C-terminal anticodon-binding domain and variable length insertions (dashed line) downstream from the active site.

AARS urzymes contain several structural modules. One module, the protozyme, appears to be older than the others [26]. Protozymes contain the ATP-binding sites in both AARS Classes. The protozymes appear to have arisen simultaneously on a single bidirectional gene [2], as outlined further below. The protozyme is at the amino terminus of Class I and at the carboxy terminus of the Class II AARS urzymes. Class I AARSs have three modules in addition to the protozyme (A)—the connecting peptide insertion within the catalytic domain, the second half of the urzyme, C, and the anticodon-binding domain, D. We settled on a nomenclature for AARS urzymes using these modular designations. Class I urzyme modules begin with the protozyme and so are designated with the three-letter amino acid name plus AC. Class II urzymes begin with the bidirectional partner or module C and end with the protozyme, so are designated (Aa)CA.

We have described urzymes derived from two Class I (TrpAC [35,36] and LeuAC [45,46]) and two Class II (HisCA [37] and GlyCA [47]) AARSs. They all speed up amino acid activation by ~10^9^-fold over the uncatalyzed rates. They speed up cognate tRNA acylation by ~10^3^-fold [38,46,47]. Further, the LeuAC urzyme actually acylates the TΨC minihelix^Leu^ about an order of magnitude faster than tRNA^Leu^ [48].

### 2.3. AARS Protozymes Are Amino Acid-Activating Catalysts That Can Be Coded by a Bidirectional Gene

A bidirectional gene encodes functional proteins from both of its strands. We made a computationally designed bidirectional gene to code 46-residue, ATP-binding subsites from Class I and II AARSs on opposite strands. Both excerpts, called protozymes, are active. In fact, catalytic proficiencies, k_cat_/K_M_, of bidirectionally coded Class I and II protozymes are the same, within experimental error, as those of wild-type TrpRS and HisRS protozyme sequences [2]. Tamura and co-workers verified our conclusions [40].

The bidirectionally coded protozymes increase in the rate of amino acid activation by more than ~10^6^-fold. That overcomes the rate-limiting step in protein synthesis [39,42]. Thus, it strongly supports participation of such genes early in the genesis of genetic coding. Moreover, the AARS protozymes were very likely to be the earliest, coded catalytic polypeptides with direct, ancestral phylogenetic relationships to the contemporary proteome.

### 2.4. Consequences of the Inverse Complementarity of Nucleic Acid Base-Pairing Duality Project Deeply into the Proteome

The coding table is exquisitely well designed to promote bidirectional coding [49,50,51]. Thirty codons—those with middle base U or A—occur as complementary codon: anticodon pairs, where one of which encodes a core and the other a surface amino acid. The amino acids related in this way define the insides and outsides of folded proteins. Water-to-cyclohexane transfer free energies, ΔG_w>c_, for Class I and II protozymes show high reflection symmetry in antiparallel alignment (Figure 3a). The tertiary structures that result from each strand are consequently anticorrelated. Surfaces that form tertiary structures (amber stripes) in one Class lie opposite amino acids that form the solvent-accessible surface (green stripes) of the other Class. Proteins translated from opposite strands of bidirectional genes thus fold up inside out from one another!

### 2.5. The Projected Duality Creates Rudimentary Nanomachinery for Chemical Free-Energy Transduction

This inversion also has a strong impact on Class-dependent AARS ligand binding (Figure 3b). Remarkably, both protozymes furnish binding determinants for both ATP and amino acid substrates. They bind ATP with higher affinity than full-length enzymes [2]. Indeed, Class I protozymes appear to be ancestral to NTP-binding sites in a broad variety of ATP- and GTPAses [52]. NMR studies by Mildvan of structurally homologous ATP-binding peptides excerpted from DNA polymerase I [53], F1 ATPase [54,55], and adenylate kinase [56,57] suggest that ATP induces folding from a largely disordered form to something resembling the structures of these peptides in X-ray crystal structures of the intact proteins. Thus, the two AARS protozymes appear to underlie the origin not only of amino acid activation, but also of NTP-linked free-energy coupling to biochemistry in general.

### 2.6. The Projected Duality Constrains Substrate Recognition by AARS Urzymes, Dividing Amino Acids and tRNA Acceptor Stems into Parallel Groups

These studies reinforce the extensive homologies of the protozymes from across the ten families of both Class I and II AARSs [26]. As is the case with the three proteins Mildvan studied, Class I and II secondary structures appear in a similar order, β–α–β. Crystal structures of AARSs complexed with analogs of their activated amino acids [58,59] (Figure 3b) reveal patterns that account for their preference for large (Class I) or small (Class II) side chains. Contacts from the central α-helix specify ATP (blue) in Class I and amino acid (salmon) in Class II protozymes. Conversely, those from the β-strands specify amino acid in Class I and ATP in Class II protozymes.

The inversion also differentiates both amino acid and tRNA acceptor stem groove recognition by AARS urzymes [60,61]. The ATP- and amino acid-binding sites put the prochiral α-phosphate into diastereroisomeric environments in Class I and II AARSs because the amino acid-binding determinants arise from opposite sides of the adenine ring plane. Consequently, there is less room for side chains in Class II AARSs [62], consistent with the uniformly smaller size of Class II amino acids.

Most Class I and II AARSs approach the tRNA acceptor stem from opposite grooves (Figure 4) [14,60,61]. Cognate tRNA binding is differentiated decisively because the 3′ DCCA terminus of Class I tRNA substrates, which approach via the minor groove, must make a sharp hairpin to enter the cognate synthetase active site. Class I urzymes promote RNA hairpin formation via specific interactions between the amino terminus of the specificity-determining helix and the phosphate group and ribose of A76. Class II tRNA substrates approach the major groove and do not require an RNA hairpin. Rather, the extended polypeptide hairpin at the C-terminus of the Class II protozyme—the Motif 2 loop—is flattened across the 1–72 base pair, fortifying the 3′-DCCA helical extension.

Both Class I and II urzymes are therefore replete with determinants not only for catalytic rate acceleration of both amino acid activation and acyl transfer to tRNA, but also for differential recognition of both amino acid and tRNA substrates. Notably, with the exception of the conserved aromatic residue—A76 ribose interaction in Class I (here mediated by Y198, Figure 4A)—these determinants appear to be rooted in a side-chain independent secondary structure. Recent combinatorial mutagenesis of full-length Class I Leucyl-tRNA synthetase and its urzyme confirmed this by showing that the native HIGH and KMSKS catalytic sequences are either non-functional or inhibitory in the LeuAC urzyme, whereas they favor catalysis of tRNA acylation synergistically in the full-length enzyme [45]. Bidirectional coding thus appears to account for the requisite primordial differentiation of both amino acids and tRNAs, hence for creating the first informational bit of the eventual genetic code.

## 3. Phylogenetics

Any account of the origin of translation must seek consistency with the historical record embedded into the proteome structural (sequence and tertiary) databases. A readout from those databases is challenging because Nature took highly convoluted pathways—assimilation of hard-to-define modular bits of genetic information, horizontal gene transfer, and mutation reversals—to elaborate the proteome. It is far easier to interpret the contemporary primary and 3D structural databases and work backward. However, the most challenging questions posed by the origins of translation, including the order in which amino acids entered the coding table, concern behavior near the oldest nodes which remain inherently ambiguous. These require supplementary assumptions.

Caetano-Anollés and colleagues [63,64,65,66,67,68,69] clarify many difficult issues posed by these databases. Their models capture salient points other investigators [70,71,72,73,74,75] fail to address. Foremost among these are the requirement for co-evolution of multiple functions [66], especially of catalysis with the interpretive machinery; the widespread sharing of modular components as the proteome diverged [64,65,66,76]; and the historical separation between architectural diversification and the adaptive radiation of genes within species [76], which parallels that between self-organization [77,78,79,80] and natural selection.

Models drawn from phylogenetics must be validated by excerpting experimental prototypes from putative nodes in phylogenetic trees and testing them for both catalytic proficiency and specificity [35]. The catalytic proficiency of the bidirectional 46-residue protozyme gene products [2,40] implies remarkable functionality in much smaller structural motifs than those examined by Caetano-Anollés. The functional granularity of the proteome’s structural patchwork has substantially higher resolution than that inferred from the SCOP database of intact protein domains [81]—which have an average length of nearly 190 residues [82].

That high-resolution mosaicity likely has crucial details about the birth of the proteome [34]. The Class I urzyme is structurally isomorphous with the TOPRIM domain [43]. Phylogenetic metrics suggest that it, too, has significant mosaic substructure, including evidence that the protozyme [26] is its oldest module. The linear dependence of the transition-state stabilization and Michaelis constant free energies on sequence length of the protozyme, urzyme, catalytic domains, and intact Class I and II AARSs (see Figure 6 in [41]) constitutes pivotal experimental evidence for the hypothesis that these represent meaningful states along the evolutionary pathway to mature AARSs.

A key goal is to define sequence probability distributions for nodes at which ancestral AARS bifurcated to form two mutually exclusive new forms. Despite the considerable promise of ancestral sequence reconstruction [83,84,85,86], however, unambiguous phylogenetic trees for both AARS superfamilies remain elusive for at least three reasons. First, specific substrate recognition required the advent of sophisticated allosteric phenomena [2,45,87,88,89,90,91]. So, the earliest genes and biological peptides were doubtless quasispecies until the proteome was almost complete. Second, the functionally relevant sequence space at the root was substantially more comprehensive than expected, in the sense that very diverse sequences had similar activity [2,45]. Third, each AARS family bifurcation to form two exclusive new letters in the coding alphabet forced Nature to decide which of the two, new amino acids worked best in every extant context, and re-equilibrate the entire extant proteome to the enlarged coding alphabet [92].

Thus, phylogenetic algorithms must be revised not only to incorporate asymmetric transition matrices [93], but also to dynamically recognize and equilibrate increases in the coding alphabet. Recent developments may improve prospects for realizing this goal [94].

Ancestral bidirectional coding [13] would have strictly coupled the sequences for specific ancestors of two AARS Classes in the earliest evolution of genetic coding. An especially important and novel role of phylogenetic analysis is to test that hypothesis by extending the work of Chandrasekaran et al. [44]. Thus, the degree of codon middle-base pairing should serve two purposes in ancestral sequence reconstructions. One would help identify the earliest pairs coupled in that way. At the same time, that would provide supporting evidence for the Rodin–Ohno hypothesis.

Experimental testing of reconstructed ancestral sequences highlights many of the challenges of other higher-order combinatorial problems. Artificial intelligence tools to aid the design of protein sequences, given a backbone scaffolding constraint have recently transformed our ability to evaluate and improve upon reconstructed ancestral sequences [95]. This capability, combined with other bioinformatic tools [96,97,98,99] affords a path for selecting sequences likely to have useful properties, simplifying a large combinatorial array to a much smaller representative sample.

As the coding specificity of the current 20 amino acid table decreases with successive ancestral nodes, the resulting ancestral sequences become increasingly like quasispecies. It becomes important to be able to characterize populations. An appropriate way to meet that challenge is to express recombinant libraries, which implies assaying populations, rather than individual species. Douglas, anticipating this requirement, has created a Bayesian estimation for the mean value and variance of the two Michaelis–Menten parameters, k_cat_ and K_M_ [100]. A key purpose of that software is to estimate the enzymatic heterogeneity of the population.

## 4. Constraints: Impedance Matching and Reciprocally Coupled Gating

Prebiotic chemistry was not arbitrary. Sutherland [18,19,20,21,22], Martin [23], and others [101,102] have identified numerous cyclic and otherwise coupled networks related to the origins of metabolism and polymer synthesis. Those networks make up the historical context in which biology emerged (see Section 1). Genetic coding marks a singular transition. That transition assembled processes governed by chemical equilibrium into computationally controlled symbolic storage and readout of amino acid chemistry. Nature then exploited that symbolic manipulation by evolving coded synthesis of life’s machinery.

The most significant obstacles Nature overcame to produce animate matter were explosive combinatorial optimization problems. The emergence of symbolic coding from random chemistry was unlikely without mechanisms that selected improbable combinations of multiple processes. Such problems preoccupied the protein-folding community [103,104,105]. In keeping with the metaphor of bootstrapping Nature’s OS, artificial intelligence neural network algorithms called constraint programming [106] provide a general approach to solving them. Such methods recently succeeded for both protein folding [96] and its inverse, protein design [95]. They succeed by incorporating a constraint surface.

Aspects of the constraint surface Nature likely navigated as it built the coding table are illustrated in Figure 5. Wills developed a rigorous equivalence between the energetic cost of errors in information transfer and the physical concept of impedance [107]. He then showed that the most efficient pathway to the coding table matched the error rates in translation to those in replication. Thus, the gradual sharpening of AARS specificity and correspondingly decreased redundancies in coding assignments should be correlated with error rates in nucleic acid replication [15,107,108]. More coding letters lead to increasingly specific AARSs in analogy to successive derailleur gears on a bicycle, so the most probable path from a binary to the contemporary 20-letter alphabet also dissipated the least chemical free energy.

Strange loops are paradoxical, cyclical, self-referential, and level-crossing feedback loops [109]. Examples abound in molecular biology [110]. The AARS•tRNA cognate pairs are perhaps the most deeply puzzling, because they interweave genetic coding with protein folding. Folding is necessary to interpret the code, and folding requires conforming to the code. The levels that cross are the folding rules and the coding rules. Strange loops are evident throughout coding, catalysis, and bioenergetics. A broader constraint is associated with recognizing that these “strange loops” [110] can be formulated generally as reciprocally coupled AND gates [111,112]. Coupling the antecedent of a second AND gate to the consequent of the first compounds the strength of the first. In this way, reciprocally coupled gates explicitly filter large numbers of inputs, greatly reducing the possibilities.

As an example, efficient coupling of ATP consumption to useful work requires both transition-state stabilization and a conformational change, while at the same time, conformational changes require both transition-state stabilization and ATP consumption (i.e., product release) [111,112]. That counterintuitive coupling illustrates how reciprocally coupled gating reduces many possibilities to few, efficiently bypassing Darwinian natural selection. As gating refines performance, the strange loop opens new possibilities. Iterative application thus contributes significantly to the emergence of order from chaos (Figure 5b).

## 5. Experimental Challenges

The coherent experimental and conceptual results described here amount to a plausible scenario by which Nature implemented genetic coding by elaborating two Classes of AARSs (Figure 6). It rests on a substantial, though partial experimental base. Deconstructing the patchwork of contemporary AARS genes, we characterized a nested hierarchy of functional modules from both AARS Classes as experimental models [35,36,37,38,43] for evolutionary intermediates Nature used to create the genetic coding language, and by implication the necessary genetic messages. Remarkably, these intermediates also trace the emergence of catalytic proficiency and the capture of chemical free energy from ATP.

The hypothesis of Rodin and Ohno that the AARS Classes originated from opposite strands of the same ancestral gene provides a plausible model for creating the first bit of coding information by providing a structural rationale for the amino acid and tRNA specificities of the earliest AARS•tRNA cognate pairs. In a qualitative sense, the structural data in Figure 3 and Figure 4 thus enhance the likelihood and Bayesian posterior probability of the bidirectional coding hypothesis substantially beyond that provided by direct experimental evidence presented in Section 2. Many details, however, remain obscure.

### 5.1. Validating the Role of Bidirectional Coding

The designed 46-residue bidirectional protozyme gene [2,40] and four Class I/II urzymes are fledgling experimental outposts in the remote terrain from which genetic coding emerged. Were such bidirectional ancestral genes and their cognate tRNAs necessary and sufficient to elaborate the full coding table? In other words, how close were they to a functional boot block for Nature’s operating system? Despite promises captured in Figure 3 and Figure 4 and evidence for their remarkable catalytic properties, there remain fundamental questions about whether and, if so how, protozyme and urzyme specificity for the two substrates became sufficient to launch and refine coded protein synthesis.

These questions fall into three categories: (a) What functionalities are possible with alphabets of a given size and composition? (b) When and how did tRNA recognition arise? (c) What factors, in addition to the growth of the coding alphabet induced successive specificity improvements? These questions and the experimental tools to answer them, especially if supplemented by enhanced phylogenetics, are considered briefly here.

How many bits (pairs of coding letters) were necessary to make bidirectional gene products sufficiently specific to achieve reflexivity? Experimental validation of reflexivity calls for designing a self-consistent alphabet and set of implementing genes. We must design functional bidirectional genes for Class I and II AARS precursors using a reduced alphabet. Then, those gene products must exhibit the experimental capability to discriminating between appropriate subsets of amino acids and tRNAs well enough to implement the corresponding alphabet.

How close have we come to this goal? Experimental LeuAC, HisCA, and GlyCA urzyme specificity spectra (see Figure 5 in [47]) discriminate against amino acids from the opposite class on average four times out of five, and all have a within-Class preference for about five amino acids. Reinforcement from tRNA groove recognition might strengthen preferences to nine times out of ten. These probabilities are consistent with urzymes administering a two-bit, four-letter alphabet as suggested in Figure 6.

Tamura’s studies [40] revealed that protozymes may discriminate poorly between different amino acid side chains, and no one has yet tested their ability to transfer the aminoacyl group to tRNA (see Section 5.2). 

Considerable phylogenetic work to establish likely sequence probability distributions and amino acid specificities at the two-bit nodes, remains before bidirectional genes can be designed to confirm such hypotheses experimentally. Even more challenging work will be required to demonstrate reflexive enforcement by protozymes of a one-bit alphabet with two kinds of letters.

2.Is a bidirectional urzyme gene feasible? Naïve analysis of the modular patchwork of Class I and II urzymes (see Figure 4A in [35]) has not resulted in an antiparallel alignment Class I and II urzyme sequences compatible with continuous bidirectional coding of their respective three-dimensional structures. That analysis likely cannot constrain protein design programs as hoped. Recently, a more suitable, alternatively threaded antiparallel alignment emerged. That alignment, also consistent with the high resolution modularity [26], may provide a template for bidirectionally coded urzymes. However, we have yet to test it.3.What limitations of bidirectional coding forced its breakdown by providing new functionality? The CP1 insertion at the C-terminus of the protozyme interrupted all extant Class I urzymes, definitively ending bidirectional coding. Eventually, CP1 significantly enhanced amino acid specificity, but only when complemented by the anticodon-binding domain [89,90]. Simultaneous acquisition of both domains seems unlikely, so one might expect a more decisive selective advantage for so significant a modular acquisition. The LeuAC urzyme converts substantial amounts of ATP to ADP in single turnover experiments [46]. If a comparable analysis of the intact catalytic domain reduced ADP production, that would suggest that CP1 initially increased the efficiency of free-energy transduction. Modular deconstruction of Class II AARSs should also shed new light.4.Can AARS urzyme acylation of TΨC minihelices confirm details of the operational code? Acylation of minihelices partially substantiated the “single-domain” model for the origin of coding [12]. Evidence that AARS urzymes catalyze acylation of full-length tRNAs [38] further strengthened that model. Recently, we showed that minihelix^Leu^ is an even better substrate for LeuAC than tRNA^Leu^ [48]. That can now enable a detailed test of the operational code.5.Can AARS protozymes catalyze tRNA acylation? Polypeptide catalysis of aminoacylation must have appeared sometime between the ancestral bidirectional protozyme gene and the emergence of urzymes. Structures illustrated in Figure 4 are quite sophisticated, even though far simpler than contemporary AARSs. As AARS protozymes likely exemplify earlier ancestral catalytic polymers, it may be notable that the motif 2 loop Class II protozymes retains much of the tRNA-binding site [61], whereas the Class I tRNA-binding site is formed largely by a helix present only in the urzyme. That asymmetry, suggesting that Class II AARSs preceded Class I AARSs, raises the profound objection that functional polypeptides must have depended minimally at least on a binary code. Ribozymes similar to the flexizyme family [113,114] might have accelerated acyl transfer from aminoacyl-5′AMP produced by Class I protozymes to proto-tRNAs, assuring provision of aminoacylated RNAs for templated protein synthesis. That would have required an ad hoc mechanism to discriminate between two types of tRNA.
6.To what extent can the elements described in Section 2.6 account for the assignments of amino acids to codons in the coding table?There are two credible models attempting to establish how Nature assembled the contemporary coding table. One [28] is driven by the need to account for the AARS Class division. The other [115] has the advantage of preserving the ability of the coding table to optimize bidirectional coding because it preserves the codon–anticodon assignments to core and surface amino acids (Figure 3a; reference [49]). Satisfying both requirements is quite difficult. Neither model appears to be consistent with the goals of the other.Our hope is that progress in ancestral reconstruction can help resolve this important dilemma. Experimental characterization will augment the specificity spectra of both both amino acid [47] and RNA minihelix substrates [48]. Such data will inevitably be necessary to identify and address the underlying questions.

### 5.2. Beyond Genetic Coding

The bidirectional gene construct suggests experimental, computational, and theoretical approaches to other questions implicit in Figure 1.

Both protozyme genes have similar distributions of conformational angles, φ and ψ, consistent with β–α–β secondary structures. Class I superfamily tertiary structures are based on the Rossmann dinucleotide-binding fold [116] and parallel β-strands interspersed with α-helices. Class II structures are based on antiparallel β-strands. Crystal structures suggest this difference is nascent in the respective protozymes. Do sequence differences between Class I and II protozymes dictate their ultimate tertiary structures. If so, how? Do they emanate from the bidirectional coding inversion (Figure 3a)? The experimental models we created to study AARS evolution may help answer these questions:(i)We can infer sequence/structure relationships from variations in both naturally occurring and designed sequence databases.(ii)Bioinformatic tools reducing tertiary structures to lower dimensions—conformational angles (φ, ψ); residue transfer free energies (ΔG_vapor>chx_, ΔG_water>chx_) [27,117,118]; TetraDA one-dimensional strings derived from Delaunay tesselation [119]; SNAPP scoring [97,98]—provide alternative multidimensional windows into structural and evolutionary determinants.(iii)The highly sensitive Malachite Green assay for phosphates generated on amino acid activation [40] affords a five-fold increase in the rate at which assays can be performed on variants of this gene.(iv)Artificial intelligence has improved both protein design [95] and structure prediction [96], creating a dynamic virtual feedback loop capable of sampling substantially larger regions of the protein sequence space expected for early nodes in AARS speciation. Pruning those sequence distributions virtually, before committing to experimental construction, expression, and testing will greatly enhance the experimental tools described above.

## 6. Did Creating the Genetic Code Require New Physics?

String theorist Edward Witten has written that “physics—like history—does not precisely repeat itself, (but) it does rhyme” [120]. This work has uncovered significant rhymes. Constraint surfaces in Figure 5 point to analogies with physical laws, notably the equivalence of the energetic cost of errors and the physical concept of impedance [107] (Figure 5a). Figure 5b draws on less obvious potential analogies that nonetheless likely conform to known physics [111]. Putting it the opposite way, impedance matching and reciprocally coupled gating provide possible mechanistic heuristics for universal extremum principles, including the minimum action principle. In that sense, investigating the genesis of animate matter may have opened new ways to view physical laws, rather than identifying a need for new physics. Biology’s secrets lie in structural and symbolic coincidences—bidirectional coding can produce two structurally distinct, but functionally similar types of AARS enzymes that antiparallel polypeptide double helices can assume identical helical parameters to those of A-form RNA [121]—that solve otherwise improbably difficult problems, rather than requiring new laws.

## Figures and Tables

**Figure 1 life-14-00199-f001:**
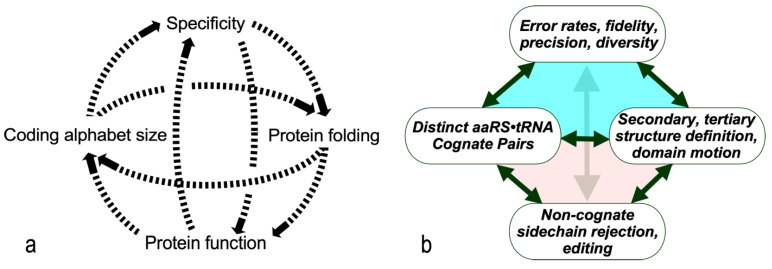
Coupled evolutionary advances favoring emergence and fostering the specialization of genetic coding. (**a**). Interdependences involved in embedding specific recognition into the sequence space of folded proteins [26]. (**b**). Experimentally measurable metrics corresponding to (**a**). Although experimental models exist for each process, confirming details requires further work.

**Figure 2 life-14-00199-f002:**
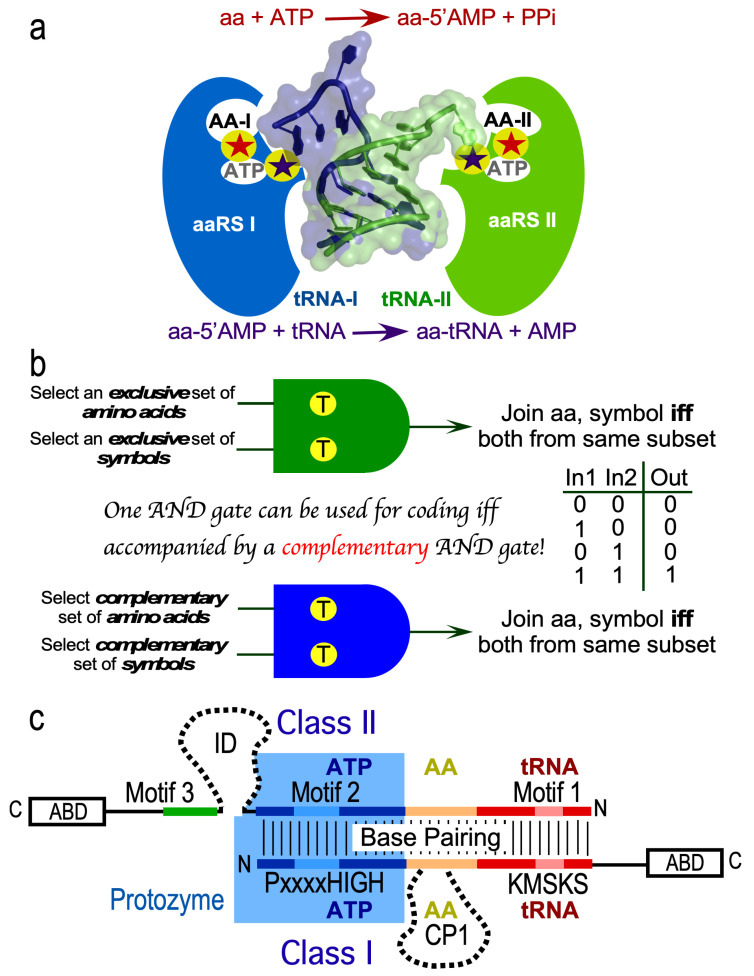
Assignment catalysis by AARSs. (**a**). Class I and II AARS•tRNA cognate pairs as envisioned by Ribas and Schimmel [11] with molecular cartoons of respective tRNA acceptor stems superimposed to highlight the opposite directions of their 3′ CCA extensions. Synthetases are shown as elliptical shapes to highlight the five distinct regions in their active site pockets. Three are cavities (white) for amino acid, ATP, and 3′ terminal adenosine. Stars represent transition-state complementarity during amino acid activation (red) and acyl transfer to cognate tRNA (purple). (**b**). AND gate pseudocode. Two inputs are detected and compared to select exclusive, complementary subsets of the two substrate groups, amino acids and tRNAs. tRNA substrates contain cognate anticodons used to read mRNA, hence are explicitly symbolic. Covalent bonds result if and only if both substrates are correct. (**c**). Schematic of bidirectional coding of Class I and II AARSs, highlighting the protozyme (blue background), the urzyme (overlapping based-paired region), and the approximate locations of major substrate-binding determinants.

**Figure 3 life-14-00199-f003:**
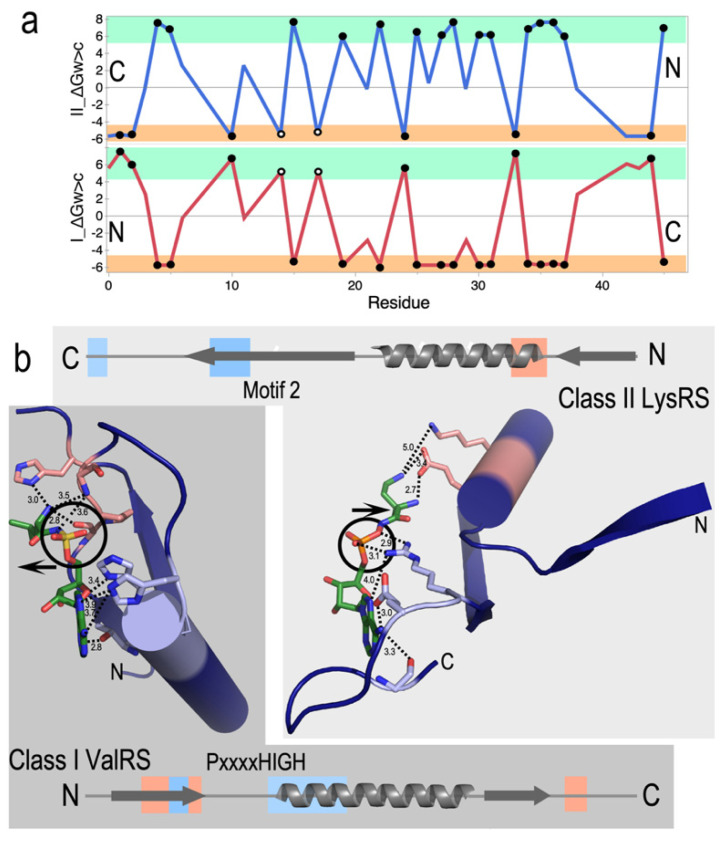
Bidirectional coding induces inside-out folding and inverted substrate binding. (**a**). Antiparallel alignment of Class I (red) and II (blue) protozymes described previously [2,40]. A significant majority (74%) of amino acid side chains have inversely related transfer free energies, ΔGw>c, from water to cyclohexane, leading to highly symmetric hydrophobicity profiles for those residues. Solid dots denote residues from restricted subsets (Ile, Val, and Leu; subclass IA) and (Asp, Lys, and Asn; subclass IIB) and account for 46% of the sequence. Open dots denote residues defining the HIGH and Motif 2 signatures. (**b**). Genetic complementarity propagates, via reflection symmetry in (**a**) into the resulting Class I and II protozyme and tertiary structures. This has functional consequences. Their succession of secondary structures is similar, but their inverted polarity means that the substrate-binding loci are inverted. ATP binds to light blue segments at the N-terminus of the α-helix in Class I (dark background) and to the C-terminus of the second β-strand in Class II (light background). Similarly, the amino acid substrate binds to salmon segments of both β-strands in Class I but to the N-terminus of the α-helix in Class II. Binding sites have the adenine ring (lower left) in approximately the same orientation to highlight the approximate stereroisomerism of the 5′ phosphate (circles). Class I amino acids point left, away from, while Class II amino acids point right, toward their protein-binding determinants (arrows).

**Figure 4 life-14-00199-f004:**
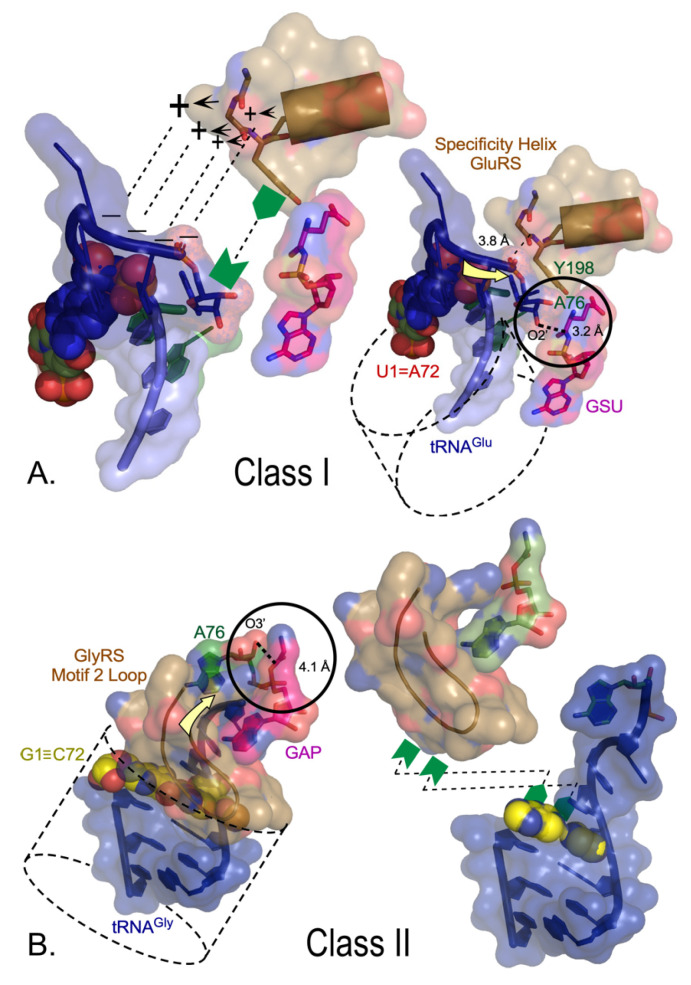
Class-dependent AARS•tRNA cognate pair formation is dictated by synthetase tertiary structure. (**A**). Class I complexes require the 3′CCA terminus to form a hairpin that is recognized by a combination of electrostatic (+, −) and aromatic/hydrophobic (green symbols) interactions with the 3′ terminal phosphate and ribose, respectively. The 3AKZ GluRS complex is typical of functionally relevant complexes of LeuRS, ArgRS, and GlnRS. (**B**). The 5E6M GlyRS complex is typical of functionally relevant complexes of Class II ThrRS, AspRS, and AlaRS. Acceptor stems are indicated by dashed cylinders in both A and B. Aminoacylation sites are circled, with A76 nucleophile-to-aminoacyl-5′AMP distance shown with a bold dashed line. The direction of the incoming CCA is indicated by yellow arrows. Interactions with cognate AARSs (sand) make very different interactions with the 1–72 base pairs (spheres). The Class I α-helix with many side chains recognizing the amino acid forms key interactions with the 3′ phosphate and the ribose moieties of A76, enforcing the characteristic hairpin turn. Those interactions do not entail the terminal base pair. In Class II complexes, the antiparallel β-hairpin of Motif 2 (brown ribbon) covers the 1–72 base pair, enforcing interactions with the helical extension of the CCA terminus [60,61] (adapted from [14]).

**Figure 5 life-14-00199-f005:**
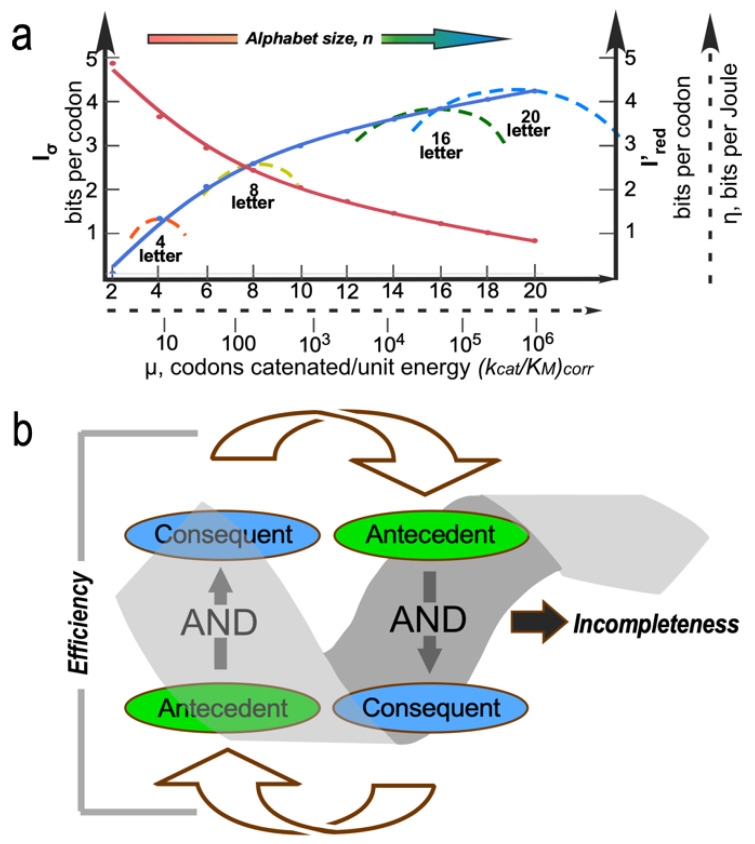
Elements of the constraint surface over which Nature optimized the coding alphabet. (**a**). Representative curve (blue) for coded information content I_σ_ (bits per codon or amino acid; left-hand Y-axis) versus alphabet size, n. Nominal rate of translation, μ, on log(k_cat_/K_M_) is indicated by the dashed proxy scale. Typical curves (dashed, color spectrum) for information transferred per unit energy expended per monomer concatenated, η, (bits per Joule per monomer, arbitrary dashed scale on right-hand Y-axis) for AARS evolution through increasing amino alphabet size, n. Redundancy (red curve), I’_red_ versus n, is averaged out over the codon and amino acid alphabets, making the most significant contribution to impedance matching. (**b**). Reciprocally coupled gating of two AND gates with the consequent of the first linked to the antecedent of the second introduces two kinds of forces. Gating efficiency is an attractive force because it concentrates inputs that satisfy both AND conditions. Coupled gates form a strange loop leading to incompleteness, hence an open-ended chemical potential-like reservoir of new possibilities.

**Figure 6 life-14-00199-f006:**
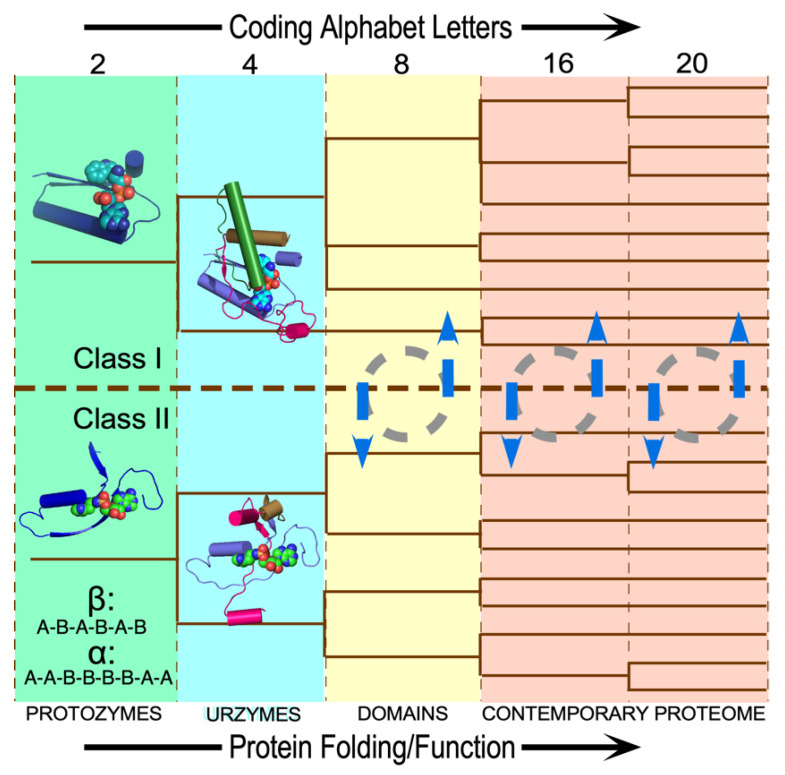
Scenario for the origin and evolution of the 20 AARSs. The size of the coding alphabet along the top defines the differently colored panels. Approximate functionality achieved for a given alphabet size is indicated along the bottom. The tree superimposed on the panels and the coding letters at any stage are to be determined. As AARS speciated, dashed circles with blue arrows denote the equilibration of functional advantages from new coding letters across the extant proteome.

## Data Availability

All data cited have been published and are available via public databases (The Protein Databank) or upon request from the author.

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
