# Peer review of "Base Pairing Promoted the Self-Organization of Genetic Coding, Catalysis, and Free-Energy Transduction"

_life, 2024, doi:10.3390/life14020199_

Round 1

Reviewer 1 Report

Comments and Suggestions for Authors

Aminoacyl-tRNA synthetases use ATP to catalyze the attachment of amino acids to their cognate tRNAs. In addition to providing the aminoacyl-tRNA substrates for protein synthesis, these enzymes provide the energy for peptide bond formation (via the aminoacyl-tRNA linkage) and are responsible for the fidelity of the process, as misacylation of tRNA results in the wrong amino acid being incorporated during protein synthesis. Based on their role in protein synthesis (i.e., translating the mRNA code into amino acids) it is postulated that the primordial aminoacyl-tRNA synthetases must have played a central role in the development of the genetic code. In this manuscript, Carter reviews efforts to elucidate the role that aminoacyl-tRNA synthetase evolution played in establishing the genetic code.

This work is based on the Rodin-Ohno hypothesis, which postulates that the mRNAs for the primordial class I and II aminoacyl-tRNA synthetases were originally transcribed from opposite strands of the same DNA sequence. To test this hypothesis, Carter and colleagues constructed truncated forms of class I and II aminoacyl-tRNA synthetases, known as urzymes, which consist of only the modules predicted by the Rodin-Ohno hypothesis. They show that these urzymes are able to catalyze both the amino acid activation and tRNA acylation reactions. They further show that the class I and II urzymes are less specific than the corresponding full-length aminoacyl-tRNA synthetases and propose that they are models for ancestral aminoacyl-tRNA synthetase forms with reduced specificity. Specifically, given their ability to preferentially recognize distinct subsets of 5-6 different amino acids, Carter postulates that the urzymes represent ancestral forms of the aminoacyl-tRNA synthetases that translated a four-letter alphabet (i.e., 4 codons). Subsequent analyses demonstrated that a more truncated form of the aminoacyl-tRNA synthetases is still active with respect to amino acid activation, albeit with further reduced specificity. Carter proposes that this truncated form (referred to as a protozyme) represents an ancestral aminoacyl-tRNA synthetase form that preceded the urzymes and translated a two-letter alphabet (i.e., 1 bit of information). By analyzing the physical models of these putative evolutionary intermediates (i.e., the protozyme and urzymes), Carter argues that their evolution is tied to both the evolution of specificity in the genetic code and the capture of chemical energy from ATP hydrolysis. This manuscript makes a good argument in support of the central hypothesis that bidirectional coding created the first informational bit of the genetic code, the implications of this hypothesis, and challenges that remain in testing of the hypothesis.

Specific Comments

1.    Page 3, lines 116-125: The paragraph begins with a systems biology approach, namely formulating the problem in terms of “AND gates”, then shifts abruptly to the idea that primordial aminoacyl:tRNA pairs used minihelices to translate the genetic code. These two ideas need to be connected more explicitly.

2.    Page 4, Figure 2: The work this figure is based on postulates that the same tRNA is recognized by class I and II aminoacyl-tRNA synthetases. How does this fit with the author’s proposed model for aminoacyl-tRNA synthetase evolution?

3.    Page 4, lines 148-161: A figure showing the relationship of the bidirectional coding (i.e., Rodin-Ohno) hypothesis to the protozyme and urzyme sequences would be helpful.

4.    Page 5, line 169: The class II aminoacyl-tRNA synthetase domain structure needs to be described.

5.    Page 7, lines 254-256: The use of the term “hairpin” in two different contexts is confusing. The text needs to clearly differentiate between the extended polypeptide hairpin loop and the hairpin at the 3’ end of tRNA.

6.    Page 10, lines 379-380: The term “strange loops” needs to be defined and the text needs to be more explicit about the proposed relationship between strange loops and the development of the genetic code.

7.    Page 13, lines 466-469: The class I protozyme lacks the determinants for tRNA acylation that are present in the class II protozyme. Are there corresponding determinants in the class I protozyme that are absent from the class II protozyme?

Comments on the Quality of English Language

None

Author Response

By analyzing the physical models of these putative evolutionary intermediates (i.e., the protozyme and urzymes), Carter argues that their evolution is tied to both the evolution of specificity in the genetic code and the capture of chemical energy from ATP hydrolysis. This manuscript makes a good argument in support of the central hypothesis that bidirectional coding created the first informational bit of the genetic code, the implications of this hypothesis, and challenges that remain in testing of the hypothesis.

I appreciate the reviewer’s succinct statement of my goals in assembling this ms.

Specific Comments

  1. Page 3, lines 116-125: The paragraph begins with a systems biology approach, namely formulating the problem in terms of “AND gates”, then shifts abruptly to the idea that primordial aminoacyl:tRNA pairs used minihelices to translate the genetic code. These two ideas need to be connected more explicitly.

The revision includes new text linking these thoughts in lines 120-135.

  1. Page 4, Figure 2: The work this figure is based on postulates that the same tRNA is recognized by class I and II aminoacyl-tRNA synthetases. How does this fit with the author’s proposed model for aminoacyl-tRNA synthetase evolution?

I suspect that the reviewer mis-interprets either Fig. 2a or the original paper by Ribas and Schimmel on which it is based. To preclude such interpretations, I added a new sentence in lines 149-151.

  1. Page 4, lines 148-161: A figure showing the relationship of the bidirectional coding (i.e., Rodin-Ohno) hypothesis to the protozyme and urzyme sequences would be helpful.

The revised Fig. 2c has such a schematic.

  1. Page 5, line 169: The class II aminoacyl-tRNA synthetase domain structure needs to be described.

Lines 180-185 have a new paragraph describing the modular architecture of Class I and II AARS.

  1. Page 7, lines 254-256: The use of the term “hairpin” in two different contexts is confusing. The text needs to clearly differentiate between the extended polypeptide hairpin loop and the hairpin at the 3’ end of tRNA.

Lines 272-280 have been rewritten to clarify the distinction between RNA and polypeptide hairpins.

  1. Page 10, lines 379-380: The term “strange loops” needs to be defined and the text needs to be more explicit about the proposed relationship between strange loops and the development of the genetic code.

A new paragraph at lines 408-417 provides both definitions and context for the discussion of strange loops.

  1. Page 13, lines 466-469: The class I protozyme lacks the determinants for tRNA acylation that are present in the class II protozyme. Are there corresponding determinants in the class I protozyme that are absent from the class II protozyme?

This is a potentially very interesting question. After reflection, however, I find I cannot address it in the terms sought by the reviewer. The direct answer seems to be no. The remarks to which the reviewer is referring seem to signal an interesting asymmetry between the Class I and II protozymes.

Reviewer 2 Report

Comments and Suggestions for Authors

The article under consideration, titled “Base Pairing Promoted the Self-Organization of Genetic Coding, Catalysis, and Free-Energy Transduction” by Charles W. Carter, Jr, provides a captivating and comprehensive exploration of the intricate processes underlying the development of genetic coding in nature. The author highlights the often-overlooked question of how nature arrived at genetic coding, emphasizing its fundamental role in understanding the evolution of life from basic biochemical components and processes.

One of the strengths of the article is its elucidation of the challenges associated with the instability of the peptide bond, shedding light on the critical role of chemical energy in driving processes such as amino acid activation and acyl-transfer. The discussion on the necessity for catalysis in these processes and the universal need for living organisms to convert free energy while coordinating cellular chemistry adds depth to the narrative. This integration of biochemical principles provides a strong foundation for readers with varying levels of expertise in the subject matter, like this reviewer.

The article's central argument, proposing that functional proteins occupy specific regions of sequence space and require a sophisticated system involving a memory of amino acid behaviour, code-keys, and a scoring function, is well-supported. The connection drawn between gene base-pairing's structural duality and the emergent links from experiments adds an intriguing layer to the discussion. The author effectively weaves together these complex concepts, making the content accessible without compromising on scientific rigor.

Furthermore, the clarity and coherence of the writing contribute significantly to the overall quality of the article. The logical progression of ideas, coupled with well-structured paragraphs, facilitates a smooth and engaging reading experience. The inclusion of pertinent examples and references to experimental evidence further bolsters the credibility of the arguments presented. In conclusion, this paper stands out as an excellent review article that significantly contributes to our understanding of the intricate processes governing genetic coding in nature. The well-articulated arguments, clear writing style, and insightful connections between various biological puzzles make it a valuable addition to the scientific literature. I recommend its publication in its present form (please only remove the various double spacing in the text).

Author Response

. The well-articulated arguments, clear writing style, and insightful connections between various biological puzzles make it a valuable addition to the scientific literature. I recommend its publication in its

present form (please only remove the various double spacing in the text).

This is a curious problem. I found the solution to that problem elusive. The double spacing would come and go, depending on where in the text it occurred. I believe I’ve found the source of, and corrected, the problem.

That said, I thank the reviewer for his/her generous evaluattion of the manuscript.

Reviewer 3 Report

Comments and Suggestions for Authors

The manuscript discusses the shared origin of Class I and Class II AARS despite their contrasting architectures. The author suggests that bidirectional coding played a crucial role in the early stages of genetic coding. The bidirectional protozymes are proposed to have increased the rate of amino acid activation, supporting their participation in the genesis of genetic coding. The inverse complementarity of nucleic acid base-pairing is discussed, projecting deeply into the proteome and creating rudimentary nanomachinery for chemical free energy transduction.

Phylogenetics is emphasized as a crucial tool for understanding the origin of translation, and the authors stress the importance of consistency with historical records embedded in proteome databases. The paper highlights the challenges posed by the origins of translation, including the ambiguity near the oldest nodes, and the need for supplementary assumptions. Questions are raised about the emergence of tRNA recognition, and the limitations and capabilities of AARS urzymes.

The paper provides a comprehensive exploration of the origin of translation, offering valuable insights and proposing plausible scenarios. However, the paper emphasizes the need for further experimental validation and more conclusive answers to the remaining challenges posed in the text, especially regarding the reflexive enforcement by protozymes of a 1-bit alphabet and the feasibility of bidirectional urzyme genes.

The reviewer suggests minor revisions.

1. To enhance reader comprehension, it is necessary to rewrite the abstract, which currently has several shortcomings. The abstract contains no information about the results obtained from the article. Only in the last sentence is there a hint of “surprising links”. What new has been done in the work, what results have been achieved, what could they reveal to the reader?

2. While the paper is centered around the Bidirectional Coding Hypothesis, it is essential for the reader to be aware that the author is, in reality, utilizing a series of interrelated hypotheses:

- Phylogenetic Models for Origin of Translation: The paper emphasizes the use of phylogenetics as a tool to understand the origin of translation. It suggests that consistency with historical records embedded in proteome databases is essential, and phylogenetic models need to be validated by experimental prototypes. Why is this hypothesis connected with the hypothesis of Bidirectional Coding?

- “How did Nature solve the “chicken-and-egg” questions by converting random chemistry into symbolic coding to interpret genes?” (Line 86), “Emergence of symbolic coding from random chemical processes was unlikely without mechanisms for selecting improbable combinations of multiple processes. Such problems preoccupied the protein folding community [93-95]” (Line 349). Why does the author uncritically use hypotheses related to "random chemistry"? The appearance of chemical substances on early Earth is an unsolved problem. Studied pathways (see, for example, the works of John Sutherland or Bill Martin on prebiotic chemistry) are varied but not arbitrary. Clearly, there was no chicken soup on early Earth, no random chemistry.

- Hypothesis about the „fist bit information“. “The elemental barrier to creating the code was creating the first bit of information.” (Line 117). This hypothesis is opposed by hypotheses about parallel competing protocodes, for example, those presented by Vetsigian, Woese, and Goldenfeld, or Nesterov-Mueller. In these approaches, information was created in parallel in competing entities. Why is it so important to start with the first bit assumption?

3. Did creating the genetic code require new physics? (Line 511)

The approach to posing the question of “new physics” seems somewhat irritating. The author himself notes that there is no need for new physical principles, but does not provide statements in the text that could justify such a question at all. The use of terms like "impedance matching" and "reciprocally coupled gating" in a biological context may be metaphorical and not necessarily directly related to the laws of physics as traditionally understood. The accumulation of hypotheses in the article, coupled with the absence of specific insights into the distribution of amino acids among codons in the genetic code, seems to have led the author to seek solace in physics. The reviewer believes this is the correct approach.

The transition to principles rooted in physics becomes imperative when unraveling the enigma of the genetic code's origin. Physics dictates that the simplest formula with minimal assumptions holds the truth for a phenomenon, regardless of its implications. Indeed, the Copernican method greatly simplified the description of the motion of the planets to minimal assumptions, although it led to the shocking conclusion that people are rushing at great speed upside down around a burning ball. Applying this principle of minimal assumptions, the combinatorial fusion cascade (doi: 10.3390/life11090975) elucidates the distribution of amino acids among codons and the emergence of the stop codons with minimal rules, shedding light on the significance of codon complementarity in forming aaRS classes (Combinatorial Fusion Rules to Describe Codon Assignment in the Standard Genetic Code. Life. 2020 doi: 10.3390/life11010004). Biochemists have already explored this concept, demonstrating peptide synthesis on complementary RNA fragments (A prebiotically plausible scenario of an RNA-peptide world. Nature. 2022 doi: 10.1038/s41586-022-04676-3.) Could the author benefit from incorporating these ideas into his experimental planning?

The reviewer acknowledges the poetic touch of "uncovering significant rhymes" (L513) in the paper. Despite the suggested corrections, the reviewer prefers not to dissect the poetic license extensively. The reviewer recommends accepting the manuscript without insisting on the proposed minor changes.

Author Response

The paper provides a comprehensive exploration of the origin of translation, offering valuable insights and proposing plausible scenarios. However, the paper emphasizes the need for further experimental validation and more conclusive answers to the remaining challenges posed in the text, especially regarding the reflexive enforcement by protozymes of a 1-bit alphabet and the feasibility of bidirectional urzyme genes.

The reviewer suggests minor revisions.

  1. To enhance reader comprehension, it is necessary to rewrite the abstract, which currently has several shortcomings. The abstract contains no information about the results obtained from the article. Only in the last sentence is there a hint of “surprising links”. What new has been done in the work, what results have been achieved, what could they reveal to the reader?

This concern is both important and relevant. I have revised the abstract to highlight what is new in the manuscript (primarily Fig. 3) and how that shapes the emphasis.

  1. While the paper is centered around the Bidirectional Coding Hypothesis, it is essential for the reader to be aware that the author is, in reality, utilizing a series of interrelated hypotheses:

- Phylogenetic Models for Origin of Translation: The paper emphasizes the use of phylogenetics as a tool to understand the origin of translation. It suggests that consistency with historical records embedded in proteome databases is essential, and phylogenetic models need to be validated by experimental prototypes. Why is this hypothesis connected with the hypothesis of Bidirectional Coding?

I am grateful for this question. The logic underlying the importance of phylogenetics is that it provides the only relic of the historical record that can be invoked in support of the bidirectional-coding hypothesis. In particular the fact that a limited analysis of published in 2013 (Chandrasekaran et al.) showed that the frequency of codon middle-base pairing in antiparallel alignments of ancestral sequences reconstructed independently from Class I and II sequence alignments. Increases as one approaches the root node. We ultimately hope to extend that approach to a full-scale analysis of all 20 AARS. Lines 366-372 address this point in a new paragraph.

-“How did Nature solve the “chicken-and-egg” questions by converting random chemistry into symbolic coding to interpret genes?” (Line 86), “Emergence of symbolic coding from random chemical processes was unlikely without mechanisms for selecting improbable combinations of multiple processes. Such problems preoccupied the protein folding community [93-95]” (Line 349). Why does the author uncritically use hypotheses related to "random chemistry"? The appearance of chemical substances on early Earth is an unsolved problem. Studied pathways (see, for example, the works of John Sutherland or Bill Martin on prebiotic chemistry) are varied but not arbitrary. Clearly, there was no chicken soup on early Earth, no random chemistry.

I agree fully with the reviewer. The original text was flawed in provoking that suggestion. Nevertheless, genetic coding does mark a singular transition from chemical processes governed by chemical equilibrium and the symbolic incorporation and exploitation of physical characteristics in coded synthesis of life’s machinery. Line 65 mentions this and lines 558-564 introduce the discussion by addressing this point.

- Hypothesis about the „first bit information“. “The elemental barrier to creating the code was creating the first bit of information.” (Line 117). This hypothesis is opposed by hypotheses about parallel competing protocodes, for example, those presented by Vetsigian, Woese, and Goldenfeld, or Nesterov-Mueller. In these approaches, information was created in parallel in competing entities. Why is it so important to start with the first bit assumption?

I am grateful also for this penetrating question. The first bit assumption seems essential to me because it provides a (putatively functional root from which subsequent coding grew. That is essentially why historical context is crucial. Notwithstanding, the “first-bit” hypothesis is consistent with simulations of competing protocodes. In fact, the work of Vestigian is not contradictory in the sense that the first bit I described necessarily led to coded peptides that are quasispecies. Lines 124-132 summarize these points.

  1. Did creating the genetic code require new physics? (Line 511) The approach to posing the question of “new physics” seems somewhat irritating. The author himself notes that there is no need for new physical principles, but does not provide statements in the text that could justify such a question at all. The use of terms like "impedance matching" and "reciprocally coupled gating" in a biological context may be metaphorical and not necessarily directly related to the laws of physics as traditionally understood.

I divided the long original paragraph, because its distinct parts require different answers. This segment of question 3 suggests that the two references to possible “new physics” are metaphorical. I welcome this suggestion, and prefer not to refute it, as a resolution of the evident difference of opinion is not evident. In fact, my intention was to suggest just the opposite—we have previously tried to make the case that both impedance-matching and reciprocally coupled gating are explicit extensions of extant physical principles. That position is clearly an opinion.

The accumulation of hypotheses in the article, coupled with the absence of specific insights into the distribution of amino acids among codons in the genetic code, seems to have led the author to seek solace in physics. The reviewer believes this is the correct approach.

I agree with both the motivation and the detailed expression of this concern. I did not attempt to describe in detail how the amino acids came to be distributed among codons. Several extant models attempt to do this, and I was unaware of the combinatorial fusion model. I have addressed this significant omission in lines 559-568 of the revision

The transition to principles rooted in physics becomes imperative when unraveling the enigma of the genetic code's origin. Physics dictates that the simplest formula with minimal assumptions holds the truth for a phenomenon, regardless of its implications. Indeed, the Copernican method greatly simplified the description of the motion of the planets to minimal assumptions, although it led to the shocking conclusion that people are rushing at great speed upside down around a burning ball. Applying this principle of minimal assumptions, the combinatorial fusion cascade (doi: 10.3390/life11090975) elucidates the distribution of amino acids among codons and the emergence of the stop codons with minimal rules, shedding light on the significance of codon complementarity in forming aaRS classes (Combinatorial Fusion Rules to Describe Codon Assignment in the Standard Genetic Code. Life. 2020 doi: 10.3390/life11010004). Biochemists have already explored this concept, demonstrating peptide synthesis on complementary RNA

fragments (A prebiotically plausible scenario of an RNA-peptide world. Nature. 2022 doi: 10.1038/s41586-022-04676-3.) Could the author benefit from incorporating these ideas into his experimental planning? The reviewer acknowledges the poetic touch of "uncovering significant rhymes" (L513) in the paper. Despite the suggested corrections, the reviewer prefers not to dissect the poetic license extensively. The reviewer recommends accepting the manuscript without insisting on the proposed minor changes.

I am most grateful for the reviewer’s graceful expression of this important, but complex and somewhat philosophical question It opens a new level of questions, following up that addressed by my previous comment.

Reviewer 4 Report

Comments and Suggestions for Authors

Journal: Life

Manuscript number: 2827271

Title: Base Pairing Promoted the Self-Organization of Genetic Coding, Catalysis, and Free-Energy Transduction

The evolution of the genetic code is tracked based on experimental models of evolutionary intermediate AARS•tRNA cognate pairs.

It starts with how the first bit of genetic information was formed, summarizes the key model for experimental study of the origin of the genetic coding table and its implementation; It deals with what can be expected from analysis of the historical record;  the constraint surface—the scoring function that probably shaped Nature’s self-organization prior to the advent of Darwinian selection is described; It surveys challenges that remain to be addressed and the experimental models with which to address them. It discusses whether understanding the emergence of animate matter required new physics.

MAJOR

In this work a splendid review of the origin and evolution of the genetic code focusing on AARS interactions with cognate tRNAs is presented. Most of the references are from the author that span a wide variety of subjects, from experiments with protozymes, urzymes, theoretical works like impedance matching and reciprocally-coupled gating, and tetraDA one-dimensional strings derived from Delaunay tessellation, until if new physics is needed to understand animate matter.

The review integrates difficult topics in the subject, and it is timely welcomed.

MINOR

Line 56:

Nature built these self-describing machines…

Nature built these self-manufacture machines…

Line 105:

“…whether or not…” replace by “whether”

Section 2. Line 110

Transfer RNAs are a like a computer programming language...

Transfer RNAs are like a computer programming language…

Figure 3.

Line 249

Please revise the phrase: “Cognate tRNA binding by s differentiated decisively…”

Line 320:

“…trees for both AARS superfamilies remain elusive for at least two reasons…”

“…trees for both AARS superfamilies remain elusive for at least three reasons…”

Figure 5a

“Codons concatenated” instead of “codons catenated”

Legend Line 363

Please clearly distinguish among n, µ, and h.

Section 6

Suggestion:

“…does not precisely repeat itself, (but) it does rhyme - the appearance of similar structures in different areas of physics, herein biology.”

Lines 521-522:

Biology’s secrets lie in structural and symbolic coincidences that solve otherwise improbably difficult problems, rather than requiring new laws.

Replace “coincidences” by “codes”.

Comments on the Quality of English Language

Author Response

Many thanks for this generous appreciation of the manuscript as a whole.

MINOR

Line 56:

Nature built these self-describing machines…

Nature built these self-manufacture machines…

Line 64 now reads “Nature evolved these self-constructing…

Line 105:

“…whether or not…” replace by “whether”

Corrected as suggested in revision

Section 2. Line 110

Transfer RNAs are a like a computer programming language...

Transfer RNAs are like a computer programming language…

Corrected as suggested in revision

Figure 3.

Line 249

Please revise the phrase: “Cognate tRNA binding by s differentiated decisively…”

The revision corrects this sentence.

Line 320:

“…trees for both AARS superfamilies remain elusive for at least two reasons…”

“…trees for both AARS superfamilies remain elusive for at least three reasons…”

The revision corrects this inconsistency.

Figure 5a

“Codons concatenated” instead of “codons catenated”

Legend Line 363

Please clearly distinguish among n, μ, and h.

This concern puzzles me. I see no ambiguity in the definitions of the three symbols, which are μ, η, which are Greek symbols, and n, which is an English character and is the alphabet size.

Section 6

Suggestion:

“…does not precisely repeat itself, (but) it does rhyme - the appearance of similar structures in

different areas of physics, herein biology.”

Again, I’m unsure what to make of this concern.

Lines 521-522:

Biology’s secrets lie in structural and symbolic coincidences that solve otherwise improbably difficult problems, rather than requiring new laws.

Replace “coincidences” by “codes”.

Codes is inappropriate in this context. Coincidences include that bidirectional genes encode two different functional proteins, that polypeptide double helices can have identical helical parameters as those in double-stranded A-form RNA, etc. The revised final sentence of the manuscript clarifies this point.